# Is the Level of Contrast Enhancement on Contrast-Enhanced Mammography (CEM) Associated with the Presence and Biological Aggressiveness of Breast Cancer?

**DOI:** 10.3390/diagnostics13040754

**Published:** 2023-02-16

**Authors:** Alaa Marzogi, Pascal A. T. Baltzer, Panagiotis Kapetas, Ruxandra I. Milos, Maria Bernathova, Thomas H. Helbich, Paola Clauser

**Affiliations:** 1Department of Medical Imaging, King Abdullah Medical City Specialist Hospital, Muzdalifah Rd, Al Mashair, Makkah 24246, Saudi Arabia; 2Department of Biomedical Imaging and Image-Guided Therapy, Division of General and Pediatric Radiology, Medical University of Vienna, 1090 Vienna, Austria; 3Department of Biomedical Imaging and Image-Guided Therapy, Division of General and Pediatric Radiology, Division of Molecular and Structural Preclinical Imaging, Medical University of Vienna, 1090 Vienna, Austria

**Keywords:** breast neoplasms, mammography, contrast media, breast cancer

## Abstract

There is limited information about whether the level of enhancement on contrast-enhanced mammography (CEM) can be used to predict malignancy. The purpose of this study was to correlate the level of enhancement with the presence of malignancy and breast cancer (BC) aggressiveness on CEM. This IRB-approved, cross-sectional, retrospective study included consecutive patients examined with CEM for unclear or suspicious findings on mammography or ultrasound. Excluded were examinations performed after biopsy or during neoadjuvant treatment for BC. Three breast radiologists who were blinded to patient data evaluated the images. The enhancement intensity was rated from 0 (no enhancement) to 3 (distinct enhancement). ROC analysis was performed. Sensitivity and negative likelihood ratio (LR-) were calculated after dichotomizing enhancement intensity as negative (0) versus positive (1–3). A total of 156 lesions (93 malignant, 63 benign) in 145 patients (mean age 59 ± 11.6 years) were included. The mean ROC curve was 0.827. Mean sensitivity was 95.4%. Mean LR- was 0.12%. Invasive cancer presented predominantly (61.8%) with distinct enhancement. A lack of enhancement was mainly observed for ductal carcinoma in situ. Stronger enhancement intensity was positively correlated with cancer aggressiveness, but the absence of enhancement should not be used to downgrade suspicious calcifications.

## 1. Introduction

Although mammography remains the standard method for breast cancer screening, its sensitivity is affected by several factors, the most relevant being breast density [1]. This limitation drives the need for adjunctive breast imaging tools able to detect breast cancer in women with dense breasts [2,3]. Until recently, contrast-enhanced magnetic resonance imaging (MRI) was the most widespread breast imaging method, offering an accurate visualization of breast vascularization [4]; this has changed with the development of contrast-enhanced mammography (CEM) [5]. Using dual-energy technology, CEM allows the evaluation of lesion vascularization [6]. This technique requires less procedural effort and is thus easier and faster to implement in the clinical workflow than MRI. Breast MRI usually requires an additional patient visit and has a number of contraindications that do not apply to CEM [7,8]. Recent publications have demonstrated that the diagnostic accuracy of CEM exceeds that of full-field digital mammography (FFDM), but the sensitivity and negative predictive value are slightly inferior to those of MRI [9]. 

It is known from breast MRI that benign and malignant lesions can be distinguished by their level of enhancement: due to higher vascularization, contrast medium uptake is earlier and more intense in malignant lesions [10]. It is also assumed that more aggressive breast cancers, characterized by high growth rates and the potential to metastasize, show stronger enhancement compared to cancers with lower proliferative activity [11]. In CEM, the strength of enhancement can be assessed by qualitative visual grading [12,13], which places the investigated lesion in contrast to normal background parenchymal enhancement. The absence of enhancement in breast MRI is a strong indicator of a benign finding and practically excludes malignancy [14,15]. Whether the same holds true in CEM has been investigated in only a few studies [16,17]. Enhancement characteristics between breast MRI and CEM differ due to a variety of reasons. Both iodine-based contrast agents and gadolinium-based contrast agents are administered intravenously and rapidly redistribute from the intravascular to the interstitial spaces [18]. This phenomenon, in the breast, is particularly visible in malignant lesions, due to the angiogenesis typical of malignant tumors [19]. Gadolinium-based contrast agents alter the relaxation time in the tissue, displaying a hyperintense signal on T1-weighted images even at low doses, particularly when using high-relaxivity contrast agents [20]. On CEM, the level of contrast is related to the attenuation of the X-ray beam, and it correlates with the concentration of the contrast in the area of interest [7,18]. Currently, the administration protocols for contrast agents in this context are widely variable and have not yet been optimized [6]. Thus, it is unclear whether the level of enhancement on CEM could be as reliable as the level of enhancement on MRI for the characterization of breast lesions [10,21].

The present study aimed to correlate the level of lesion enhancement on CEM with the presence of malignancy and breast cancer (BC) aggressiveness. 

## 2. Methods and Materials

### 2.1. Patient Population

The local ethics committee approved this retrospective study, and the need for informed consent was waived. In this retrospective, single-center, cross-sectional study, the images of consecutive patients who underwent CEM from July 2018 to December 2020 were evaluated. Inclusion criteria were women who had inconclusive or suspicious breast lesions found on screening or diagnostic mammography and/or handheld breast ultrasound (BI-RADS 0, 3, 4 or 5). Exclusion criteria included the absence of a reference standard; ongoing neoadjuvant chemotherapy; and the presence of a known and already biopsied malignancy. The standard of reference was histology obtained with image-guided biopsy in all lesions, excluding lesions with uncertain malignant potential identified at core needle biopsy. Lesions of uncertain malignant potential (B3) were included only if surgical excision was available, and surgical histology was considered as the reference standard in these cases. 

### 2.2. Contrast-Enhanced Mammography 

The system used to perform the examinations was a Mammomat Revelation unit (Siemens, Erlangen, Germany). A dual-energy examination, consisting of high-energy (HE; 49 kVp) and low-energy (LE; 26–32 kVp) images, was obtained consecutively during a single breast compression. A non-ionic iodine contrast medium (Iomeron^®^ 400, Bracco, Milan, Italy) was administered at 1 mL/kg body weight and at a rate of 3 mL/s with a power injector (Ulrich Medical, Ulm, Germany). Contrast injection was followed by a 20 mL saline flush. Image acquisition began 90–120 s after the injection of contrast medium. The exam was carried out as follows: craniocaudal (CC) of the affected side, CC of the contralateral side, medio-lateral oblique (MLO) of the affected side, MLO of the contralateral side. The generation of subtracted CEM images was done via weighted subtraction using a fully automatic, locally adjusted, tissue-thickness-dependent subtraction factor. 

### 2.3. Image Analysis 

The images were evaluated by three breast radiologists (R1, R2, R3) with eight, 10, and 14 years of experience, respectively, in breast imaging and at least one year of experience with CEM, and they were blinded to patient clinical data and histopathology results. Readings were performed on dedicated workstations in separate sessions for each reader. Readers were asked to evaluate breast density on low-energy (LE) images, according to the ACR BI-RADS 5th edition [22]. Then, they assessed the presence of lesions. Mass, calcifications, or asymmetric densities were evaluated on LE and then on high-energy (HE) images for the corresponding CEM enhancement intensity. The enhancement intensity was evaluated in a semi-quantitative manner using a scale from 0 to 3, with 0 representing no enhancement (absent), 1 subtle enhancement, 2 moderate enhancement, and 3 distinct enhancement. 

### 2.4. Histology and Molecular Analysis

Suspicious lesions were biopsied using the standard image-guided core needle or vacuum-assisted technique. Histopathology diagnosis was performed by experienced breast pathologists according to the WHO guidelines, which provided the B classification [23]. Immunohistochemical staining against estrogen/progesterone receptor (ER/PR), human epidermal growth factor receptor 2 (HER2), and ki67 (proliferation rate, MIB-1) was distinguished [24].

### 2.5. Statistical Analysis

Statistical analysis was performed using SPSS 23.0 (SPSS, IBM) and Med-Calc (MedCalc Software Ltd., online version). Inter-rater agreement was assessed using a quadratic-weighted Kappa test, which was interpreted as follows: values ≤ 0 indicated no agreement and 0.01–0.20 slight, 0.21–0.40 fair, 0.41–0.60 moderate, 0.61–0.80 substantial, and 0.81–1.00 almost perfect agreement. 

The diagnostic performance of the enhancement scores was assessed by calculating the area under the receiver operating characteristic (ROC) curve. Sensitivity, specificity, positive predictive value (PPV), and negative predictive value (NPV), as well as positive and negative likelihood ratios (LR+, LR-), were calculated after dichotomizing grade 0 enhancement as negative and grade 1, 2, and 3 enhancement as a positive test result, with 95% confidence intervals (95%CI) calculated for all the measurements. False-negative cases were defined as a mass or a calcification without any enhancement (grade 0). Fagan’s nomogram was used to estimate the pre-test probability at which the post-test probability of a negative CEM result (absence of enhancement) would meet BI-RADS 3 benchmarks (2% malignancy rate). The chi-square test was used to perform cross-tabulation data collection to compare the interpretations of all readers to the histology result. The comparison of enhancement intensity grades between invasive and non-invasive lesions (benign, B3, and DCIS) was analyzed using a one-way ANOVA test. Spearman’s rank-order correlation (*Rs)* was run to examine the relationship between enhancement intensity grades and immunohistochemical results (ER, PR, HER2, and MIB-1/ki-67). A *p* value of < 0.05 was considered statistically significant.

## 3. Results

### 3.1. Lesion Characteristics 

A flowchart of the cases is shown in Figure 1. A total of 156 lesions (93 malignant (59%); 63 benign (41%)) in 145 patients (mean age: SD of 59 ± 11.6 years; range: 87-31 years) were included. Breast density was distributed as follows: 13 (8.3%) category A, 71 (45.5%) category B, 62 (39.7%) category C, and 10 (6.4%) category D. The majority of malignant breast lesions were invasive ductal carcinoma (IDC) (*n* = 71, 76.3%); twelve lesions (19%) were classified as lesions of uncertain malignant potential (B3) at biopsy and were confirmed benign at surgery; thus, these were included in the benign histology subgroup (*n* = 63). Detailed histology and molecular subtypes are presented in Table 1.

### 3.2. Diagnostic Performance and Inter-Reader Agreement

The ROC curve revealed good diagnostic performance using enhancement intensity grades to distinguish between malignant and benign lesions; the area under the ROC curve ranged from 0.801 to 0.844, with a mean value of 0.83 (Figure 2). There were no significant differences in diagnostic performance between readers (R1–R2, *p* = 0.198; R1–R3, *p* = 0.319; R2–R3, *p* = 0.791). 

Mean sensitivity was 95.4% and NPV was 82.2% using a cut-off for the enhancement intensity grade > 0 (Table 2). Using this cutoff, the mean negative likelihood ratio was 0.12. This value allowed the potential exclusion of breast malignancy, achieving BI-RADS 3 benchmarks of a less than 2% malignancy rate up to pre-test probabilities of 18%. Using the cutoff of > 0, mean specificity was 35.4% and PPV was 68.5% (Table 2). 

Inter-reader agreement for grading the enhancement intensity ranged from substantial (0.76 between R1 and R2, 0.79 between R1 and R3) to almost perfect (0.87 between R2 and R3).

### 3.3. Comparison of Enhancement Intensity, Histological Results, and Molecular Subtype

Invasive cancer presented predominantly with distinct (range: 52.4–70.7%, mean: 61.8%) or moderate enhancement (range: 22.0–31.7%, mean: 27.7%) (Figure 3). Two invasive cancers with no enhancement were considered false-negative (FN): one case was recorded by two readers, and the other case was recorded by one reader. The DCIS enhancement intensity patterns were similar to those of benign lesions on CEM, with six of 11 cases of DCIS (54.5%) reported as absent enhancement: one by all readers, four by two readers, and one by one reader (Figure 4). Most of the lesions with absent enhancement were of the luminal B type. No triple-negative-type lesion was found with absent enhancement.

In clinical practice, both FN cases of invasive cancer and DCIS manifested with suspicious microcalcifications and, therefore, were not missed on LE mammography and CEM. 

Benign lesions presented more commonly with absent (mean benign: 34% and B3: 38.9%) or subtle enhancement (mean benign: 31.4% and B3: 43.1%) than malignant lesions. Benign lesions that more often presented with moderate or marked enhancement were fibroadenomas, adenosis, mastitis, and fat necrosis. 

There was a significant difference in the enhancement scales between the invasive and non-invasive lesions (*p* < 0.001). 

There were positive and significant correlations between enhancement intensity scales by all readers and immunohistochemistry markers (ER, PR, and MIB-1/ki-67) (*p* < 0.001). In contrast, there was no correlation between the grade of enhancement and HER2 receptor status (*p* < 0.868).

## 4. Discussion

The results of our study showed that most breast cancers present with significant enhancement on CEM. The sensitivity of enhancement for breast cancer detection on CEM was 95.4%, if any enhancement was considered suspicious. However, every third cancer showed no or only subtle enhancement, suggesting that the absence of enhancement should not be used to downgrade suspicious lesions on low-energy images, and particularly not to exclude malignancy in the presence of calcifications, as ductal carcinoma in situ often did not show any enhancement. 

A higher intensity of contrast enhancement was moderately correlated with surrogates of cancer aggressiveness. Low-enhancing malignant lesions were more often in situ carcinomas and slow-growing, hormonal-receptor-positive lesions.

Since its introduction, CEM has been promoted as an alternative for contrast-enhanced breast MRI [7,8]. While initial studies have reported comparable sensitivity and specificity for CEM and breast MRI, the ability to rule out breast cancer has not been specifically investigated [9]. We used Fagan’s nomogram to define a pre-test probability up to which a negative CEM would lead to post-test probabilities below the BI-RADS 3 benchmarks (2% malignancy rate) using the negative likelihood ratio calculated in our study, thus evaluating contrast only as method with which to define malignancy. Our results show an averaged LR- of 0.12. According to Fagan’s nomogram, lesions with a pre-test probability of 18% or less, i.e., BI-RADS 4a lesions, could be downgraded to BI-RADS 3, and biopsy could be avoided.

In our analysis, we achieved very high sensitivity when considering every enhancement as suspicious for malignancy, even subtle enhancement. The high sensitivity of CEM, however, comes at a price, as the simple presence of enhancement is non-specific. The specificity of CEM can be significantly improved by the evaluation of the low-energy images. The presence of a mass enhancement with a correlate on pre-contrast images had the highest positive predictive value for malignancy, while non-mass enhancement and foci with no correlate on low-energy images were rarely associated with malignancy [25]. To improve the diagnostic performance, both in terms of sensitivity and specificity, morphological information from low-energy images should always be interpreted in association with the functional information from the recombined images [8,16,26]. We were able to demonstrate that the level of enhancement can also be useful for lesion characterization: a stronger enhancement was associated with a higher likelihood of malignancy. The value of the association between morphological and functional data, and thus the additional information from enhancement characteristics, has long been recognized in contrast-enhanced breast MRI when the enhancement level is evaluated in combination with morphological findings [10,26]. 

We found an overlap in the enhancement of benign and malignant lesions. While most benign lesions presented with no or subtle enhancement, some benign findings, such as adenosis, fibroadenomas, or mastitis, presented in some cases with a moderate or marked enhancement [27]. However, some malignant lesions presented with no or subtle enhancement—in particular, DCIS. Among the false-negative cases, two of 82 (2.4%) invasive cancers showed an absence of enhancement, while six of 11 (54.5%) DCIS showed this feature. These results are not novel, as the lack of detectable enhancement in DCIS has already been described in the literature [9,28,29]. Particularly in this case, the absence of enhancement when suspicious calcifications are visible should not be used to exclude malignancy. Suspicious mammographic calcifications should undergo biopsy, independent of the presence of enhancement on CEM [16,28]. Previous studies have indicated that a negative MRI in the presence of calcifications, with a low level of suspicion, could be used to exclude malignancy [15]. Our results suggest that the negative predictive value of absent enhancement on CEM is lower than that of breast MRI [9]. In addition, it must be stressed that subtle enhancements should also be called positive to maximize the NPV of a negative CEM exam.

Moderate and distinct enhancement (grade 2 or 3) was more often found in malignant lesions, particularly in invasive cancers. In general, a higher intensity of enhancement on CEM is associated with a higher probability of malignancy. Similar results were also found by Łuczyńska et al. [16] and Rudnicki et al. [12]. In their analysis, they found that a qualitative and quantitative evaluation of lesion enhancement can be used to improve the diagnostic performance of CEM, particularly for less experienced readers. In a preliminary work, the same group also showed a good correlation between the enhancement curve type in MRI and the level of enhancement on CEM [30]. Deng et al. also showed that a quantitative evaluation can aid in distinguishing benign from malignant enhancement [17]. Nicosia et al. [31] included the information about the level of enhancement, together with other internal characteristics of the enhancement, in a score designed to improve diagnostic performance when reporting CEM. 

We were also able to demonstrate that more aggressive molecular subtypes, such as triple-negative breast cancers, present more often with distinct enhancement, while slow-growing, hormonal-receptor-positive cancers often present with moderate or subtle enhancement. Bicchierai et al. also showed that luminal B cancers are more often false negative on CEM, as compared to other tumor types [28]. To date, very few monocentric studies have focused on the correlation between the level of enhancement on CEM and breast cancer aggressiveness [32,33]. Łuczyńska et al. [32] measured enhancement quantitatively and reported higher contrast levels for luminal than for non-luminal cancers. These results are not in line with our findings. We hypothesize that the patient selection and methods of evaluation of the enhancement might have determined these differences. This underlines the need for further, multicentric studies on the topic. In their review of the current evidence, Vasselli et al. [33] also showed that luminal carcinomas usually show enhancement, in line with the results from Łuczyńska et al. [32], but also confirmed the correlation between the proliferation rate and level of enhancement, as shown by our results. In our analysis, the only biomarker that did not correlate with the level of enhancement was Her2 status. This is also not in agreement with other published studies [28,32], underscoring the importance of larger analyses.

As mentioned, some of the studies that evaluated the level of lesion enhancement on CEM used not only qualitative, but also quantitative methods. The application of quantitative methods can be helpful in improving inter-reader agreement and reducing variability in the interpretation. We found substantial to almost-perfect inter-rater agreement for grading enhancement intensity, showing that even a semi-quantitative or qualitative assessment can be easily applicable for different readers in CEM. This result is in line with the current evidence [12].

Our study has several limitations, one of which is the retrospective approach, which is associated with the possibility of selection bias. This was alleviated by the consecutive patient recruitment: the results thus reflect the diagnostic performance of CEM in the population where it is actually applied. The non-quantitative visual enhancement intensity grading approach by which lesion enhancement was assessed is another limitation; however, there is no established quantitative approach for enhancement on CEM, and considering the high agreement between readers, our approach can be considered robust and easy to implement in clinical practice. The intensity of enhancement could be affected by the amount of contrast administered, breast thickness, and the compression force applied. These last two factors were not considered in this retrospective analysis, and their effect on the enhancement level, therefore, remains elusive. A previous study discussed the impact of breast compression during MRI biopsy on the level of enhancement [34]. However, it remains unclear whether these findings can be directly translated to CEM.

## 5. Conclusions

The presence of enhancement on CEM has high sensitivity for breast cancer. Breast cancers with higher enhancement intensity tend to be biologically more aggressive. In case of a lack of enhancement, malignancy can be ruled out up to pre-test probabilities of 18%, potentially obviating the need for biopsies in BI-RADS 4a lesions, particularly in the absence of suspicious mammographic calcifications. The absence of CEM enhancement does not exclude malignancy in cases of mammographic microcalcifications. 

## Figures and Tables

**Figure 1 diagnostics-13-00754-f001:**
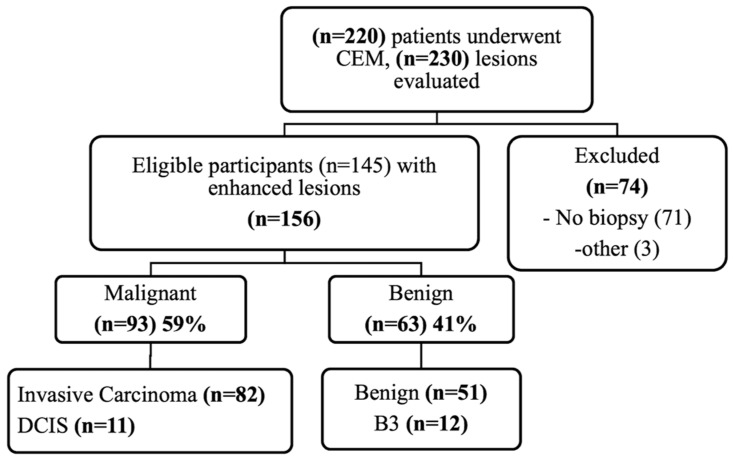
Flowchart describing the patient population. CEM: contrast-enhanced mammography; DCIS: ductal carcinoma in situ; B3: lesions of uncertain malignant potential.

**Figure 2 diagnostics-13-00754-f002:**
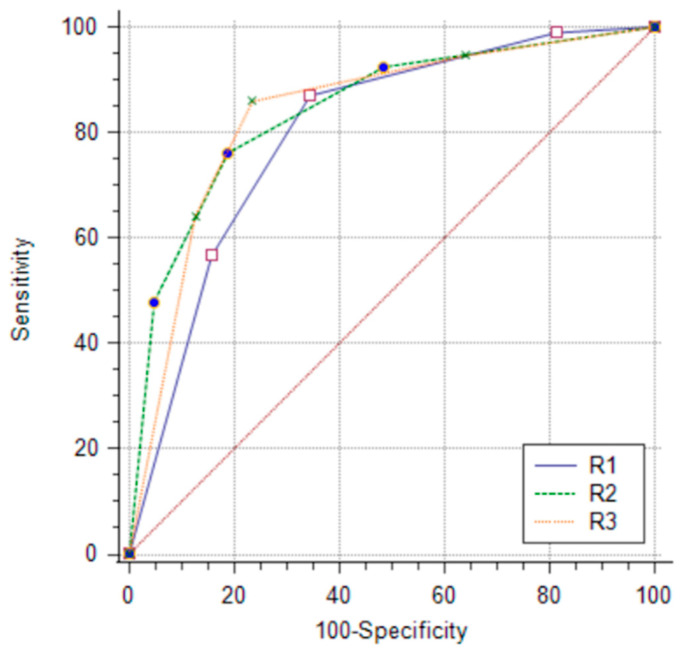
Comparison of ROC curves.

**Figure 3 diagnostics-13-00754-f003:**
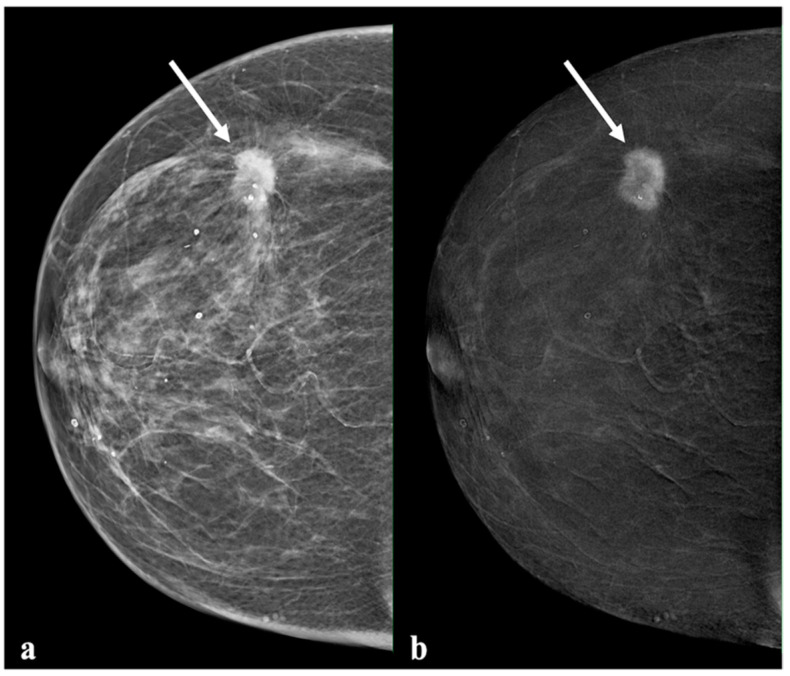
Example of a lesion with marked enhancement (Grade 3). A 71-year-old patient with a spiculated lesion in the outer quadrant of the right breast. Histology showed an invasive cancer. Mammography ((**a**), low energy) showed a dense, irregular, and spiculated mass (arrow). (**b**) The recombined image (**b**) showed minimal background parenchymal enhancement and a mass with a marked rim enhancement (arrow).

**Figure 4 diagnostics-13-00754-f004:**
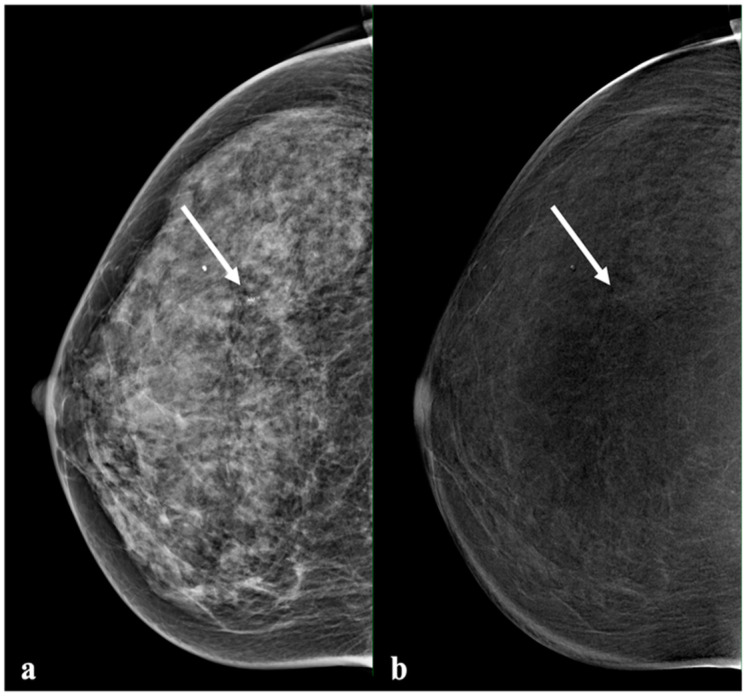
Example of a lesion with no enhancement. A 55-year-old patient with a small group of suspicious calcifications in the right breast. Histology showed high-grade DCIS. Mammography ((**a**), low energy) showed a group of pleomorphic calcifications with a linear distribution (arrow). The recombined image (**b**) showed a mild background parenchymal enhancement and suspicious enhancement in the area with the calcifications (arrow).

**Table 1 diagnostics-13-00754-t001:** Final histological characteristics of biopsied lesions.

				Average of Enhancement Intensity Grade (%)			
	Subtype		*n (%)*	Absent	Subtle	Moderate	Distinct
Benign			63/156 (40%)	34.0%	37.9%	15.7%	12.4%
	Adenosis, sclerosing adenosis		6/63 (9%)				
	Fibroadenoma, fibroepithelial hyperplasia		11/63 (17%)				
	Benign epithelial proliferation		14/63 (22%)				
	Benign breast tissue, Pseudoangiomatous stromal hyperplasia		3/63 (5%)				
	Inflammation		17/63 (27%)				
	**B3**		12 /63 (19%)	38.9%	44.5%	11.1%	5.5%
	Atypical lobular hyperplasia		4/12 (33%)				
	Papillary lesion		5/12 (42%)				
	Flat epithelial atypia		3/12 (25%)				
Malignant			93/156 (60%)				
	**DCIS**		11/93 (11.8%)	36.4%	36.4%	18.2%	9.1%
	Luminal A type		3/11 (27.3%)				
	Luminal B type		6/11 (54.5%)				
	HER2 type		2/11 (18.2%)				
	Triple-negative		0/ 11 (0%)				
	**Invasive carcinoma**		82/93 (88.2%)	1.2%	9.3%	27.7%	61.8%
	**Invasive ductal carcinoma**		71/93 (76.3%)				
	Luminal A type		14/71 (19.7%)				
	Luminal B type		35/71 (49.3%)				
	HER2 type		5/71 (7%)				
	Triple-negative		17/71 (24%)				
	**Invasive lobular carcinoma**		7/93 (7.5%)				
	Luminal A type		1/7 (14.3%)				
	Luminal B type		6/7 (85.7%)				
	**Mucinous carcinoma**		1/93 (1%)				
	Luminal A		1/1 (1%)				
	**Papillary carcinoma**		3/93 (3.2%)				
	Luminal B type		2/3 (66.7%)				
	Triple-negative		1/3 (33.3%)				

B3: lesion of uncertain malignant potential; DCIS: ductal carcinoma in situ; HER2: human epidermal growth factor receptor 2. Corresponding average of enhancement intensity grades.

**Table 2 diagnostics-13-00754-t002:** Diagnostic performance of CEM enhancement intensity for all readers.

	Sensitivity	95% CI	Specificity	95% CI	PPV%	NPV%	+LR	−LR
**Reader 1**								
>0	98.9	94.1–100	18.8	10.1–30.5	63.6	84.6	1.2	0.06
>1	86.9	78.3–93.1	65.6	52.7–77.1	78.4	86.9	2.5	0.20
>2	56.5	45.8–66.8	84.4	73.1–92.2	83.9	56.4	3.6	052
**Reader 2**								
>0	92.4	84.9–96.9	51.6	38.7–64.2	73.3	80.0	1.9	0.15
>1	76.1	66.1–84.4	81.3	69.5–89.9	85.4	70.3	1.06	0.29
>2	47.8	37.3–58.5	95.3	86.9–99.0	93.6	55.0	10.2	0.55
**Reader 3**								
>0	94.6	87.8–98.2	35.9	24.3–48.9	68.7	82.1	1.5	0.15
>1	85.9	77.0–92.3	76.7	64.3–86.2	84.0	77.4	3.7	0.18
>2	64.2	53.5–73.9	87.5	76.8–94.4	88.1	61.8	5.2	0.41
**All readers**	95.4	84.9–100	35.4	10.1–64.2	68.5	82.2	1.5	0.12

## Data Availability

Not applicable.

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
