# Peer review of "Is the Level of Contrast Enhancement on Contrast-Enhanced Mammography (CEM) Associated with the Presence and Biological Aggressiveness of Breast Cancer?"

_diagnostics, 2023, doi:10.3390/diagnostics13040754_

Round 1
Reviewer 1 Report
The authors present a novel topic not widely treated on the literature. CEM is a promising technique which use is widely implemented in the clinical practice. Degree of enhancement has not been investigated in the current literature but it could be a useful tool to assist the breast radiologist in daily clinical practice. The paper is well presented and of easy reading.
My only comment is that in pag 2 line 58 the phrase is not clear and would benefit of reformulation 'used relatively less sensitive'
Author Response
R: My only comment is that in pag 2 line 58 the phrase is not clear and would benefit of reformulation 'used relatively less sensitive'
A: We thank the reviewer for the positive comments. We also agree that the sentence is not very clear. We have reformulated the sentence to increase its clarity.
Reviewer 2 Report
Thank you for requesting to provide a review of this article, which has a subject of high interest.
The main purpose of the analysis was to correlate the level of enhancement with the presence of malignancy and breast cancer aggressiveness on CEM (contrast-enhanced mammography).
The main question adressed in the research was whether there is an association of the strength of enhancement on CEM with malignant diagnosis and cancer aggressiveness, so the researchers conducted an IRB approved, cross-sectional, retrospective study.
The study is a retrospective single-center cross-sectional analysis during a period of time between July 2018 to December 2020. The topic is original and relevant in the field and brings usefull knowledge regarding the subject. A comprehensive search strategy was used and so, the images were evaluated by 3 breast radiologists with 8, 10 and 14 years of experience in breast imaging and at least 1 year of experience with CEM. One of the main reasons to add contrast enhanced imaging to conventional mammography and ultrasound was to detect otherwise occult cancer. The review methodology was comprehensive with screening and data extraction. When it comes to the methodology used, no specific improvements should be considered from my point of view.
The conclusions are consistent with the evidence and the arguments presented, and they adress properly to the main question which conducted the analysis.
The references are appropriate and well suited for this kind of study.
Regarding the figures and pictures used in the article, they provide suitable information about the cases and show significant statistical references. They are also well understandable and the information is easy to be followed. There are no other comments required about these items, from my point of view.
Regarding the structure and accuracy of the phrases, the manuscript has well structured information, with supported evidence and well structured phrases.
The manuscript is original and well defined. The results provide an advance in current knowledge. The results are being interpreted appropriately and are significant, as well as the conclusions.
The article is written in an appropriate way.
The study is correctly designed and the analysis is being performed at high standards, so the data are robust enough to draw the conclusion.
Surely the paper will attract a wide readership.
The English language is appropriate and well understandable.
There are a few things to add in the lines below, but the article should be published after the corrections are made:
Line 24: patients, not „patient”
Line 69: the absence, not „absence”
Line 69: standard reference, not „standard of reference”
Line 71: the standard reference, not „the standard of reference”
Line 93: Then, they, not „They then”
Author Response
We would like to thank the reviewer for the very positive comments.
R: There are a few things to add in the lines below, but the article should be published after the corrections are made:
R: Line 24: patients, not „patient”
A: Corrected.
R: Line 69: the absence, not „absence”
A: Corrected.
R: Line 69: standard reference, not „standard of reference”
A: Corrected.
R: Line 71: the standard reference, not „the standard of reference”
A: Corrected.
R: Line 93: Then, they, not „They then”
A: Corrected.